# Influence of Nitrogen Application Rate on the Importance of NO_3_^−^-N and NH_4_^+^-N Transfer via Extramycelia of Arbuscular Mycorrhiza to Tomato with Expression of *LeNRT2.3* and *LeAMT1.1*

**DOI:** 10.3390/plants12020314

**Published:** 2023-01-09

**Authors:** Xiaocan Xie, Zhe Huang, Weixing Lv, Houteng Zhu, Guoming Hui, Ronghua Li, Xihong Lei, Zhifang Li

**Affiliations:** 1Beijing Key Laboratory of Growth and Developmental Regulation for Protected Vegetable Crops, Department of Vegetable Science, College of Horticulture, China Agricultural University (CAU), Haidian District, Yuanmingyuanxilu 2, Beijing 100193, China; 2Beijing Agricultural Extention Station, Huixinxili 10, Changyang District, Beijing 100029, China

**Keywords:** arbuscular mycorrhizal fungi, nitrogen transfer, tomato, *LeAMT1.1*, *LeNRT2.3*

## Abstract

Arbuscular mycorrhizal fungi (AMF) form mutualistic symbiotic relationships with many land plants and play a key role in nitrogen (N) acquisition. NO_3_^−^-N and NH_4_^+^-N are the main sources of soil mineral N, but how extraradical mycelial transfer affects the different N forms and levels available to tomato plants is not clear. In the present study, we set up hyphal compartments (HCs) to study the efficiency of N transfer from the extramycelium to tomato plants treated with different N forms and levels of fertilization. Labeled ^15^NO_3_^−^-N or ^15^NH_4_^+^-N was placed in hyphal compartments under high and low N application levels. ^15^N accumulation in shoots and the expression of *LeNRT2.3*, *LeAMT1.1*, and *LeAMT1.2* in the roots of tomato were measured. According to our results, both ^15^NO_3_^−^-N and ^15^NH_4_^+^-N were transported via extraradical mycelia to the shoots of plants. ^15^N accumulation in shoots was similar, regardless of the N form, while a higher ^15^N concentration was found in shoots with low N application. Compared with the control, inoculation with AMF significantly increased the expression of *LeAMT1.1* under high N and *LeNRT2.3* under low N. The expression of *LeAMT1.1* under high N was significantly increased when NO_3_^—^N was added, while the expression of *LeNRT2.3* was significantly increased when NH_4_^+^-N was added under low N. Taken together, our results suggest that the N transfer by extraradical mycelia is crucial for the acquisition of both NO_3_^−^-N and NH_4_^+^-N by the tomato plant; however, partial N accumulation in plant tissue is more important with N deficiency compared with a higher N supply. The expression of N transporters was influenced by both the form and level of N supply.

## 1. Introduction

Arbuscular mycorrhizal fungi (AMF) can form symbiotic relationships with more than 80% of land plants and play a key role in their nutrition [1,2]. After symbiosis is established, mycorrhizal roots with a “mycorrhizal nutrient absorption pathway” improve the mineral nutrient content of plants via a hyphal network known as the extramycelium (ERM), which is an extension of the plant root system [3,4]. Plants promote this symbiosis, as they are commonly limited by one of the two major nutrients, phosphorus (P) and nitrogen (N) [5]. Under such conditions, the mycorrhizal roots have two means of nutrient absorption: the plant pathway and the mycorrhizal pathway. The plant pathway involves the absorption of nutrients through high- or low-affinity absorption transporters in the epidermis or root hairs. For nutrients with low mobility in the soil, absorption through the plant pathway is usually limited by their depletion in the zone around the root. In contrast, the mycorrhizal pathway involves high-affinity nutrient transporters in the ERM, which take up nutrients and transport them along the hyphae to the rhizosphere hyphae (IRM) in the root cortex [1]. 

As one of the most important macronutrients, N accounts for 1–5% of the dry weight of plants. Over the last two decades, it has also been recognized that AMF plays a crucial role in the uptake by plants [6], while the soil nitrogen level is one of the factors affecting the inoculation effect of AMF [7]. The absorption of plant N transporters is induced by mycorrhization [1]. Isotopic labeling with ^15^N directly demonstrates that AMF hyphae can absorb and transport mineral N from the soil to their host plants [8]. As AMF can obtain enriched N sources, N transfer from hyphae to hosts may be huge [9]. However, compared with root absorption, AMF-derived N alone may be limited, as the potential N absorption and transport rates of mycelia are only higher than those of roots with low N (both NO_3_^−^ or NH_4_^+^) content in their soil [10].

Both the amounts and forms of N in the cultivation medium can affect the mycorrhizal infection rate, the amount of N transported by AMF to host plants, and mycelial density [11]. In most soil environments, the main form of mineral N is NO_3_^−^; however, in wetlands or highly acidic soils, NH_4_^+^ is dominant [12]. Although both forms of N (NO_3_^−^ or NH_4_^+^) can be absorbed by the external hyphae of AMF (*Rhizophagus intraradices*) and transported to the host plant [13,14,15,16], the hyphae preferentially absorb NH_4_^+^ [14,17,18]. The amount of N transferred to plants is high following an NH_4_^+^ fertilizer application [19]. However, applying only NH_4_^+^ reduces the activity of mycorrhiza compared with applying only NO_3_^−^ [20,21].

Mycorrhizal formation can directly affect the process of plant nutrient absorption and metabolism; further, it can make the growth and development of plants more advantageous than non-mycorrhizal plants and improve crop yield and fruit quality [22]. The tomato plant (*Lycopersicum esculentum* L.) is the second most important vegetable crop worldwide; however, the effect of AMF on N uptake by tomato plants in relation to N availability and forms has yet to be identified. 

At the molecular level, both the NH_4_^+^ and NO_3_^−^ transporters of hosts are regulated in AMF symbiotic plants [23,24]. In tomatoes, the NH_4_^+^ transporters of *LeAMT1.1* and *LeAMT1.2* are expressed in root hairs and leaves under N-deficient conditions, while under hydroponic growing conditions, the transcript level of *LeAMT1.2* in the roots increases after NO_3_^−^ or NH_4_^+^ supplementation, whereas that of *LeAMT1.1* is induced by N deficiency [25]. AMF excrete NH_4_^+^ to levels that can be sensed by tomato roots, and this is consistent with the induced expression of *LeAMT1.2* by as little as ≥1 µM external NH_4_^+^ with root-associated N2-fixing bacteria [26]. Rice plants colonized by *R. irregularis* strongly induce the expression of an NH_4_^+^ transporter (*OsAMT3.1*) in roots under both low and high rates of N application. As AMF increases NH_4_^+^, AMT expression could be changed due to colonization.

However, as mycorrhizal-inducing N transporters are up-regulated, the expression of nitrate transporter genes changes in host plants, thus changing the ability of plants to obtain NO_3_^−^ [24,27,28,29]. Hildebrandt et al. (2002) found that inoculation with *R. irregularis* up-regulated *LeNRT2* in tomato roots [30]; in particular, *LeNRT2.3* is related to mycorrhization and is abundantly expressed in root cells containing AMF structures, such as plexus branches and vesicles [30]. These results indicate that AMF colonization positively affects nitrate uptake from the soil and nitrate allocation to the plant partner, probably preferentially mediated by *LeNRT2.3*. So, *LeNRT2.3* functions as a low-affinity transporter, whose activity allows higher N-use efficiency in tomatoes [31]; therefore, AMF colonization positively affects nitrate uptake from the soil and nitrate allocation to the plant partner, probably preferentially mediated by *LeNRT2.3* [30].

How different N levels available and N forms in the mycorrhizal symbiosis system induce the expression of these nitrogen transporters is not fully understood. We hypothesized that N transport via AMF hyphae and *LeNRT2.3*, *LeAMT1.1,* and *1.2* expression might be correlated with N status and N forms in the hyphosphere. In the present study, hyphal compartments were used to explore the effects of two N levels and forms on mycorrhizal tomato plants. The colonization rates, plant nutrition, growth status, and *LeNRT2.3*, *LeAMT1.1*, and *1.2* expression were monitored. 

## 2. Results

### 2.1. Effects of AMF on Nitrogen Uptake by AMF and ^15^N in Shoots

In Table 1, the data show that ^15^N abundances were detected (from 0.074‰ to 0.138‰) in all mycorrhizal plants with all N forms and levels. The results were significantly higher in mycorrhizal plants with NH_4_^+^ fertilization in HCs under lower-N fertilization (Table 1). There was a significant difference between N levels, from 078‰ to 0.118‰ of ^15^N in shoots, which, with the high-N fertilization of plants, was 20.4 µg plant-1, compared with low N, which was 16.3 µg per plant (Table 1).

With high-N fertilization, neither the N concentration nor the content per plant was affected by AMF inoculation, while under low N application, the N concentration was increased from 1.24% to 1.46%, and N uptake was increased from 114.4 mg to 140.1 mg per plant. The P concentration and uptake were not affected by the difference in N fertilization (Table 2). The N form added in HCs did not affect the N absorbed by plants (Table 2). Furthermore, the colonization levels of AMF and hyphal length were similar among N levels and forms. However, the ratio of the colonization rate to the hyphal length was 2.9 with NO_3_^−^ and 4.5 with NH4+ in HCs (Table 1).

Neither shoot nor root biomass was significantly affected by inoculation with AMF and did not significantly differ between mycorrhizal plants and non-inoculated plants (Table 3). There was a large difference owing to the N fertilization levels. The total biomass of tomato plants was 19.1–19.8 g plant^−1^ with high-N fertilization, while with low-N fertilization, it was 10.2–10.7 g plant^−1^. With AMF inoculation, the biomass was not significantly affected compared with uninoculated plants (Table 3). With low N application, both the shoot and total biomass were slightly increased owing to the NO_3_^−^ added in HCs compared with NH_4_^+^ (Table 3).

### 2.2. Effects of AMF on Nitrogen Transporter Expression

Transcript levels of the *LeAMT1.1* and *LeNRT2.3* genes were induced in the roots of mycorrhizal plants, and this differed among N levels and forms in HCs (Table 4). Inoculation with AMF (*Funneliformis mosseae)* significantly up-regulated the expression of *LeAMT1.1* and *LeNRT2.3* genes with high- and low-N fertilization, respectively. Significant up-regulation of *LeAMT1.1* was observed with NO_3_^−^, while that of *LeNRT2.3* was associated with NH_4_^+^ in HCs (Table 2).

## 3. Materials and Methods

### 3.1. Experimental Site and Design

#### Experimental Protocol and Treatments

The substrate used was 3–5 mm of sterilized vermiculite with two compartments: one was the root compartment (RT), and the other was the hyphal compartment (HC) in each 3.5 L pot with an air gap separating the RT and HC. The extraradical mycelium (ERM) in fungal HCs had a volume of 250 mL surrounded by a 30 μm mesh membrane through which hyphae but not roots could grow, and each RT had a volume of 2.5 L. The ^15^N transfer between the colonized root compartment and the mycelial compartment only takes place via fungi: i.e., both diffusion and mass flow between the inner compartment and the pot were prevented due to the air gap between the RT and HC compartments [16,32]. As listed in Table 5, two levels and two forms of N fertilization were applied in the root compartments (root + fungal pots) and in HCs. NH_4_^+^ and NO_3_^−^ were added as (^15^NH_4_)_2_SO_4_ and K^15^NO_3_. All HCs with treatments of NH_4_^+^ were supplemented with 1.2 mg of nitropyridine to prevent nitrification. The plants were fertilized with a nutrient solution modified following Hoagland and Arnon (1950) that supports the growth of tomato plants [33]. The substrates (per liter) comprised 160 mg of N and 100 mg of Ca added as KNO_3_ and Ca (NO_3_)_2_∙4H_2_O, 55 mg of P as KH_2_PO_4_, 220 mg of K and 65 mg of S as K_2_SO_4_, 50 mg of Mg as MgSO_4_∙7H_2_O, 10.4 mg of Fe as Fe-EDTA, 10 mg of Zn as ZnSO_4_ 7H_2_O, and 10 mg of Cu as CuSO_4_∙5H_2_O based on dry substrates. Water was supplied based on the loss of weight according to the growth requirements of the plants. Due to mycorrhizal colonization resulting in the advantage of P nutrition, P fertilization in AMF plants was not a limiting factor. 

In each pot, 50 g of the AM inoculum *Funneliformis mosseae* was added as sand containing 10 spores per gram to the AM treatments, while for the non-AM treatments, the same amounts of sterilized inoculum and washed microorganisms and nutrition were added to each control part. Sterilization of non-AM inoculums involved heating the inoculums to 121 °C for 20 min. The inoculums were filtered through 5–8 μm filter paper with distilled water to collect possible microorganisms, and the microorganism wash was added in no-AMF treatments [16]. The non-mycorrhizal plants had N in addition to HCs, but due to the air gap, there was no nutritional connection between the root compartments and HCs. Fifty grams of AM fungal inoculum was mixed into the respective potted matrix, containing 9 spores per gram of inoculum. Control plants (non-AM) received the same amount of autoclaved (121 °C, 20 min) inoculum.

There were eight treatments, and each treatment had four replicates (*n* = 4), with a total of 64 pots. The potted tomato plants were randomly placed on the seedbed. The position of the basin was rotated once a day to eliminate the influence of the placement position on the growth and weight, and the plants were watered every day. The temperature in the greenhouse was controlled at 23 ± 2/15 ± 2 °C day/night, and the light density was increased using lamps. Plants were harvested after an experimental period of 7 weeks. 

The roots were extracted and washed, and the fresh weight was measured. Subsamples of fresh root material were taken to analyze AMF root colonization. Both shoots and subsamples of roots were dried at 65 °C, and the biomass was measured. 

Shoot N and P concentrations were measured using a DC plasma Echelle Spectrometer (Beckman Instruments) and the Kjeldahl digestion method. The root colonization rate was determined according to a modified method of Phillips and Hayman (1970) [34]. ^15^N abundance was measured by an isoprene precisION isotope ratio mass spectrometer (IRMS). The average ^15^N abundance of standard sample acetanilide (*n* = 3) measured by the instrument was subtracted from the original data to obtain the ^15^N abundance of tomato shoots.

### 3.2. Hyphal Length in HCs

Referring to the vacuum pump microporous suction filtration method [35], 10 g of matrix sample was weighed in the mycelial chamber, 250 mL of deionized water was added and stirred in the agitator for 2 min, and then the sample was mixed evenly for more than 40 min with a 30 μM mesh screen so that the screen surface was slightly inclined. The suspension was passed through the stacked sieve, and the sample was gently rinsed with deionized water. The sieved product remaining on the screen surface was washed with 250 mL of deionized water, stirred quickly for 1 min, and then left to stand for 1 min. A vacuum pump was used for vacuum suction filtration. A 5 mL volume of 0.05% trypan blue solution was added to the filter membrane and soaked for 5 min; the vacuum pump was turned on to drain the dye, and the filter membrane was clamped with tweezers and placed on a slide. The stained filter membrane was placed under a 200X microscope for observation, 10 visual fields were counted, and the number of cross points between extramycelial hyphae and the grid was recorded; the length of the extraradical mycelium was determined [36].

### 3.3. Root Sampling and Relative Expression of Transporters

Tomato root samples for RNA extraction were taken randomly from each subplot a few days before harvesting the whole plants and were stored at −80 °C. Total RNA was isolated from these root tissues using a “Quick RNA Isolation Kit” (Huayueyang Biotechnology Co., Ltd., Beijing, China), after which their cDNA was synthesized using “FastKing cDNA Dispelling RT SuperMix” (Tiangen Biotech Co., Ltd., Beijing, China). The resulting RT reaction product was used as a template for real-time quantitative polymerase chain reaction (RT-qPCR) analysis. RT-qPCR was run on a QuantStudioTM 6 Flex System, for which the primers were designed in Primer Premier 6 software, and all amplicons were between 80 and 200 nucleotides in size. The specific primer sequences were as follows:5′-CCGCCGCTTCATACATCTGCAA (forward),5′-GCGAAACCAAGCTGCATGGAGA (reverse) for *LeAMT1.1*;5′-TTCCCTCATCTCGGCAGCTCAG (forward),5′-CCGCGTAGGTGGTGTTTGTGAG (reverse) for *LeAMT1.2;*5′-GGGCTACTACACTTCCTCTGG (forward), 5′- CCTCCAGCTCCTGTCATACC (reverse) for *LeNRT2.3*; 5′-TCGTAAGGAGTGCCCTAATGCTGA (forward), 5′- CAATCGCCTCCAGCCTTGTTGTAA (reverse) for *LeUBI* [37].

The obtained product RNA extractions were used as a template for RT qPCR analysis. Real-time fluorescence quantitative PCR was performed on a Quantstudiotm 6 Flex System with “TB green”^®^ Premix Ex Taq™. RT qPCR analysis was carried out with kit II (Baoriyi Biotechnology Beijing Co., Ltd. Beijing, China), and the reaction system was 10 µL, including 5 µL of Taq™, 2.0 µL of dye II, 0.2 µL of positive and negative specific primers, 3 µL of cDNA template, and 1.4 µL of ddH_2_O. The RT qPCR procedure involved a reaction at 95 ℃ for 30 s. The comparative threshold cycle method of ΔΔCt was adopted to quantify and analyze the relative RNA expression levels. The Ct values of the target genes imported by the system were normalized to the Ct values of ubiquitin by applying the following equation: ΔCt = Ct target − Ct housekeeping. The fold change was calculated from the equation 2^−ΔΔCt^, where ΔΔCt = ΔCt sample – ΔCt Control [38].

### 3.4. Statistical Analysis

The data were recorded in MS Excel sheets and analyzed using IBM SPSS 20.0 software to determine mean values and standard errors. The statistical results derived from the experiment were expressed as means ± SE. Differences among the means were analyzed via a one-way ANOVA followed by Fisher’s least significant difference (LSD) for the multiple comparison test (*p* ≤ 0.05) to determine whether significant differences existed between plants inoculated with AMF strains and the uninoculated control. Univariate analysis of variance was also performed to analyze the main effects observed for the AMF strains and the control sample. We did not compare the statistical differences in data between two growing seasons, owing to the use of completely different cultivars; one produced large fruit, and the other produced small fruit. Regression analysis used ANOVA, as regression was virtually identical to the underlying models. The test statistic *F* was used to test for the significance of the regression model. Multiple coefficients of determination R^2^ were used to test the overall effectiveness of the entire set of independent variables. In explaining the dependent variable, its interpretation was similar to that for simple linear regression: the percentage of variation in the dependent variable that was collectively explained by all of the independent variables.

## 4. Discussion

### 4.1. Nitrogen Transport and Acquisition via AMF with N Levels and Forms in HCs

Nitrogen acquisition in plant tissues was significantly correlated with N fertilizer application levels and AMF inoculation under conditions of low N application (Table 2). The concentrations of ^15^N binding were from 0.074‰ to 0.138‰ in the shoot tissues of all mycorrhizal plants; with high N application, ^15^N binding (0.078‰) was lower than that with low N application (0.118‰) (Table 1). However, the total ^15^N transported via the extramycelium to shoot parts showed no significant difference between N levels, even with 20.4 µg per plant with high levels of N, compared with 16.3 µg at low levels of N (Table 1). Under high N application, there were no differences in either the N concentration or N content between mycorrhizal and non-inoculated plants (Table 2). In contrast, with low N application, the N concentration was increased by mycorrhization by 17.7% (from 1.24% to 1.46%), almost to the same level as plants with high N application. The N uptake by plants was 22.5%, increasing from 114.4 mg to 140.1 mg per plant, owing to the double effects of biomass and concentration.

Although ^15^N binding was not significantly different between NH_4_^+^ and NO_3_^−^ applications in HCs, the actual ^15^N transfer was 14.2 µg per plant with NO_3_^−^ application and 18.4 µg with NH_4_^+^ with low N application (Table 3). This difference implies that more ^15^NH_4_^+^ was transported from HCs to host plants via the hyphae compared with NO_3_^−^ (Table 1). This difference had no further effects on biomass accumulation in tomato plants; however, the biomass was increased when NO_3_^−^ was added to HCs (Table 3). These results suggest that almost the same amount of N transfer via MP, in the case of NO_3_^−^, had a greater influence on biomass accumulation as compared to that of NH_4_^+^ supplied to the extramycelium. With NO_3_^−^ in HCs, P uptake was significantly increased by 11.3% as the result of higher biomass at the low-N fertilization level (Table 3). It is reported that AMF contributes substantially to the N nutrition of their host plants [6]. Hyphae can directly and effectively utilize inorganic compounds and transfer a large amount of N to the roots of host plants [19,39]. The direct labeling of ^15^N has shown the flux of N through AM fungal hyphae to plants (*Andropogon gerardii*) [40]. 

In the present study, no differences in N uptake were shown with the two levels of N application in the HCs; however, ^15^N binding was higher with low N application than with high N application. This demonstrated that N transfer from the fungus to the host plant was similar at high and low levels of N application; however, with a lower N application, the HP becomes more important than with higher N rates. These results indicate that mycorrhization plays a substantial role in the absorption of plants regardless of N availability (Table 2 and Table 3). Under lower N availability, the mycorrhizal pathway becomes more important compared with the root pathway. A similar result has been reported, showing that high amounts of N application can significantly decrease N uptake by mycorrhizal plants from the soil [15]. When nutrients were insufficient, the advantages of mycorrhizal symbionts were reflected because the nutrients absorbed by plant roots were insufficient to support normal growth, while sufficient nutrients often inhibited the infection of fungi in the root system of host plants [5]. In summary, it may be concluded that a substantial amount of N can be adsorbed and transported from fungi to their host plants, and only the N uptake by hyphae, i.e., the hyphae pathway related to the root pathway, is influenced by the interaction of the N nutritional status in the environment of both the roots and fungi. This agrees with the previous hypothesis that the hyphae of AMF may absorb NH_4_^+^ preferentially over NO_3_^−^, but that the export of N from the hyphae to the roots and shoots may depend on the amount of N supplied/available for uptake [41]. However, in the present study, increased growth was not accompanied by greater concentrations of N and P in the shoots of plants. Taking biomass into account, the total content of P in shoots was increased. 

### 4.2. Transporter Genes LeAMT1.1, LeAMT1.2, and LeNRT2.3 Were Regulated by Inoculation with AMF in the Root Tissue of Tomato Plants

As previously reported, the expression of the encoded *LeNRT2.3* protein is related to AMF colonization [30]. In our study, the expression of *LeNRT2.3* in roots was significantly increased following inoculation with AMF compared with the control plants at low N levels (Table 4). Although it is a low-affinity transporter, a difference in expression was not detected between the two N levels (Table 4). *LeNRT2.3* expression was not correlated with the N form with high N application but had a significantly higher expression level with NH4+ compared with that of NO_3_^−^ with N deficiency (Table 4). As N is a major limiting factor for plant growth and yield, genes affect plant growth through nitrate uptake or remobilization. The higher expression levels of *LeNRT2.3* in flowers and leaves indicate that *LeNRT2.3* plays a pivotal role in shoot development [31]. *LeNRT2.3* is also suggested to play a key role in the xylem transport of N from roots to shoots and in N uptake by roots [31]. Taken together, the expression of *LeNRT2.3* driven by symbiosis could be important for N-use efficiency in tomatoes, and its induced expression indicates a higher N-use efficiency in tomatoes [42].

The expression of only *LeNRT2*.*3* among the transporters assayed was higher in AMF-colonized tomato roots than in non-colonized controls. AMF colonization caused the significant expression of a nitrate reductase gene of *G. intraradices*. The results may mean that AMF colonization positively affected nitrate uptake from the soil and nitrate allocation to the plant partner, probably preferentially mediated by *LeNRT2*.*3*. In addition, part of the nitrate taken up is reduced by the fungal partner itself and, if in excess, may then be transferred as glutamine to the symbiotic plant partner [30]. The expression of *LeNRT2*.*3* is negatively controlled by ammonium but, remarkably, not by glutamine [30]. 

The specific expression of these up-regulated AMT genes in arbuscule-colonized cortical root cells has been shown in *M. truncatula* [29], *L. japonicus* [28], *G. max* [24], and *S. bicolor* [43]. In the present study, regarding the two important high-affinity NH_4_^+^ transporters in roots, *LeAMT1.1* was up-regulated by inoculation with AMF, especially with NO_3_^−^ feed in HCs with high N application, while there were no significant differences in LeAMT1.2 between treatments (Table 4). Other research work has reported strong inductions of *LeAMT1.1* and *LeAMT1.2* gene expression in mycorrhizal roots, evidence that host plants had NH_4_^+^ transporters that were up-regulated under AMF colonization, with the specific expression of the up-regulated AMTs genes in arbuscule-colonized cortical root cells shown in *M. truncatula* [29], *L. japonicus* [28], *G. max* [24], and *S. bicolor* [43]. In particular, AMF symbiosis down-regulated *OsAMT1.1* expression under low-N conditions (1.825 mM NO_3_^−^) but not under high-N (7.5 mM NO_3_^−^) conditions [44]. In the present study, *LeAMT1.1* was significantly increased by inoculation with AMF and high N application and particularly up-regulated by the addition of NO_3_^−^ in HCs (Table 4). 

*LeAMT1.1* and *LeAMT1.2* are differentially regulated by N and contribute to root-hair-mediated NH_4_^+^ acquisition from the rhizosphere; the transcript levels of *LeAMT1.2* increased after NH_4_^+^ or NO3± application, while *LeAMT1.1* was induced by N deficiency [45]. *LeAMT1.2*, an important high-affinity NH_4_^+^ transporter, was reported to have contrasting responses to *LeAMT1.1* and was induced by N application [45]. By contrast, in the present study, the expression of *LeAMT1.2* was affected by neither mycorrhization nor the N level or form (Table 4). *LeAMT1.2* mRNA levels are controlled in a concentration-dependent manner by the NH_4_^+^ concentration or the plant N status, and peak expression occurs at 2–5 µM NH_4_^+^, with a further increase in NH_4_^+^ causing a reduction [26]. In our previous study, the expression of *LeAMT1.2* was significantly induced by AMF inoculation in an isolation-specific manner [46]. 

As N is a major factor determining plant growth and yield, it likely influences plant growth by modulating N uptake rates or remobilization activity [31]. The induction of N transporters varied with the level of N application and the N form in HCs; however, their increasing expression indicated a higher N-use efficiency in tomatoes. This plays a key role in the xylem transport of nitrate from roots to shoots and uptake in roots [31]. In AMF symbiosis, several studies indicate that plants absorb a large amount of N through the mycorrhizal pathway [12,19]. In the present study, AMF hyphae absorbed and transported both nitrate and ammonium N to the shoots of tomato plants with both high and low levels of N application, while under low N levels, the transported N became more important with a higher N application rate, although almost the same amount of N was transported via extraradical mycelia. Inoculation with AMF significantly increased the expression of *LeNRT2.3* and *LeAMT1.1*, which was also related to the N level and form in hyphal compartments. In conclusion, substantial amounts of both NO_3_^−^-N and NH_4_^+^-N can be transferred via extramycelia to their tomato hosts with the colonization of AMF. Under a low N supply in root environments, the partially transferred N in the plant’s total N uptake is more important than under high N supply. The expression of *LeAMT1.1* and *LeNRT2.3* were differentially influenced due to N supply levels.

## Figures and Tables

**Table 1 plants-12-00314-t001:** Tomato shoot ^15^N abundance, different N levels and forms on tomato root colonization, and hyphal length of the fungal compartment after inoculation with AMF.

Treatments		^15^N Abundance	^15^N Transported	AMF Colonization	Hyphal Density
		(‰)	(μg·plant^−1^)	(%)	(cm g^−1^ Substrate)
HN ^#^	NO_3_^−^	0.074 ± 0.006 b	20.7 ± 2.13 a	58.3 ± 5.18 a	22.3 ± 4.70 a
	NH_4_^+^	0.083 ± 0.014 b	20.0 ± 4.00 a	61.7 ± 1.65 a	10.5 ± 1.97 a
LN ^#^	NO_3_^−^	0.097 ± 0.011 ab	14.2 ± 1.32 a	53.3 ± 4.09 a	16.2 ± 5.19 a
	NH_4_^+^	0.138 ± 0.015 a	18.4 ± 2.50 a	58.3 ± 3.18 a	16.1 ± 3.35 a
**N levels** ^&^					
HN		0.078 ± 0.021 a	20.4 ± 2.11 a	60.0 ± 2.59 a	16.4 ± 3.23 a
LN		0.118 ± 0.033 b	16.3 ± 1.53 a	55.8 ± 2.58 a	16.1 ± 2.86 a
**N forms in HCs** ^&^					
NO_3_^−^		0.086 ± 0.021 a	17.5 ± 1.69 a	55.8 ± 3.20 a	19.2 ± 3.44 a
NH_4_^+^		0.110 ± 0.040 a	19.2 ± 2.21 a	60.0 ± 1.77 a	13.3 ± 2.08 a
**Significance**					
N levels		**	ns	ns	ns
N forms		ns	ns	ns	ns
N levels * N forms		*	ns	ns	ns

Note: HN means high-nitrogen treatment (160 mg L^−1^) in the root compartment, and LN means low-nitrogen treatment (94mg L^−1^) in the root compartment. HCs are hyphal compartments. #: The results are mean ± SE (*n* = 4). The error line is the standard error. &: The results are mean ± SD, *n* = 8. Multiple comparisons were performed by two-way ANOVA, and different letters indicate a significant difference between treatments (*p* < 0.05). The same letter indicates no significant difference between treatments. *, *p* < 0.05; **, *p* < 0.01; ns, non-significant, same as below.

**Table 2 plants-12-00314-t002:** Effects of AMF and N form on tomato shoot N and P under different N levels.

Treatments	HN				LN			
	N Concentration	N Uptake	P Concentration	P Uptake	N Concentration	N Uptake	P Concentration	P Uptake
	(%)	(mg plant^−1^)	(mg plant^−1^)	(mg plant^−1^)	(%)	(mg plant^−1^)	(mg plant^−1^)	(mg plant^−1^)
Inoculation								
+AMF	1.46 ± 0.10 a	263.7 ± 23.4 a	3.49 ± 0.23 a	62.55 ± 1.88 a	1.46 ± 0.03 a	140.1 ± 4.76 a	4.17 ± 0.06 a	40.0 ± 1.08 a
−AMF	1.49 ± 0.02 a	259.3 ± 5.68 a	3.31 ± 0.07 a	57.85 ± 4.23 a	1.24 ± 0.02 b	114.4 ± 1.86 b	4.24 ± 0.06 a	39.3 ± 0.75 a
Nitrogen forms in HCs								
AMF HCs NO_3_^−^	1.58 ± 0.19 a	289.6 ± 44.0 a	3.54 ± 0.37 a	64.4 ± 7.67 a	1.45 ± 0.03 a	147.58 ± 6.66 a	4.16 ± 0.11 a	42.2 ± 1.40 a
AMF HCs NH_4_^+^	1.34 ± 0.01 a	237.8 ± 13.0 a	3.45 ± 0.33 a	60.7 ± 4.75 a	1.46 ± 0.06 a	132.67 ± 4.93 a	4.18 ± 0.07 a	37.9 ± 0.73 b
Significance								
AMF	ns	ns	ns	ns	***	***	ns	ns
Nitrogen forms	ns	ns	ns	ns	ns	ns	ns	ns
AMF * Nitrogen forms	ns	ns	ns	ns	ns	ns	ns	*

Note: The results are mean ± SD, *n* = 8. Multiple comparisons were performed by two-way ANOVA, and the same letter indicates no significant difference between treatments. *, *p* < 0.05; ***, *p* < 0.001.

**Table 3 plants-12-00314-t003:** Effects of AMF and N form on tomato plant biomass under different N levels.

Treatments	HN			LN		
	Shoots	Roots	Total Plant	Shoots	Roots	Total Plant
	(g plant^−1^)	(g plant^−1^)	(g plant^−1^)	(g plant^−1^)	(g plant^−1^)	(g plant^−1^)
Inoculation						
+AMF	17.9 ± 0.61 a	1.88 ± 0.09 a	19.8 ± 0.61 a	9.61 ± 0.29 a	1.12 ± 0.06 a	10.7 ± 0.03 a
−AMF	17.5 ± 0.33 a	1.68 ± 0.14 a	19.1 ± 0.26 a	9.25 ± 0.08 a	1.01 ± 0.06 a	10.3 ± 0.12 a
N forms in HC						
NO_3_^−^	18.1 ± 0.67 a	1.89 ± 0.09 a	20.0 ± 0.74 a	10.20 ± 0.36 a	1.05 ± 0.09 a	11.2 ± 0.35 a
NH_4_^+^	17.8 ± 1.11 a	1.86 ± 0.18 a	19.6 ± 1.06 a	9.08 ± 0.27 b	1.20 ± 0.06 a	10.3 ± 0.32 b
Significance						
±AMF	ns	ns	ns	ns	ns	ns
N forms	ns	ns	ns	ns	ns	ns
AMF *N forms	ns	ns	ns	*	ns	ns

Note: HN means high-nitrogen treatment (160mg L^−1^) in the root compartment, and LN means low-nitrogen treatment (94mg L^−1^) in the root compartment. HCs are hyphal compartments. The results are mean ± SD, *n* = 8. Error line is the standard error. Multiple comparisons were performed by two-way ANOVA, and different letters indicate a significant difference between treatments (*p* < 0.05). The same letter indicates no significant difference between treatments. *, *p* < 0.05.

**Table 4 plants-12-00314-t004:** Effects of AMF and N form on nitrogen transporter genes expression of tomato root under different N levels.

Treatments	HN			LN		
	*LeNRT2.3*	*LeAMT1.1*	*LeAMT1.2*	*LeNRT2.3*	*LeAMT1.1*	*LeAMT1.2*
Inoculation						
+AMF	1.12 ± 0.29 a	1.88 ± 0.44 a	1.26 ± 0.17 a	1.12 ± 0.23 a	0.72 ± 0.28 a	0.95 ± 0.32 a
−AMF	0.58 ± 0.21 a	0.83 ± 0.29 b	1.11 ± 0.17 a	0.51 ± 0.12 b	1.30 ± 0.22 a	1.10 ± 0.16 a
Nitrogen forms (HCs)						
NO_3_^−^	0.97 ± 0.29 a	2.67 ± 0.56 a	1.36 ± 0.36 a	0.68 ± 0.17 b	0.33 ± 0.03 a	0.73 ± 0.09 a
NH_4_^+^	1.27 ± 0.56 a	1.09 ± 0.14 b	1.16 ± 0.10 a	1.56 ± 0.23 a	1.10 ± 0.50 a	1.18 ± 0.68 a
Significance						
±AMF	ns	**	ns	*	ns	ns
Nitrogen forms (HCs)	ns	*	ns	ns	ns	ns
±AMF * Nitrogen forms	ns	*	ns	*	ns	ns

Note: HN means high-nitrogen treatment (160mg L^−1^) in the root compartment, and LN means low-nitrogen treatment (94mg L^−1^) in the root compartment. HCs are hyphal compartments. The results are mean ± SD, *n* = 8. Error line is the standard error. Multiple comparisons were performed by two-way ANOVA, and different letters indicate a significant difference between treatments (*p* < 0.05). The same letter indicates no significant difference between treatments. *, *p* < 0.05; **, *p* < 0.01.

**Table 5 plants-12-00314-t005:** Treatments of root compartments and hyphal compartments.

	Treatments	Root Compartment (RC)		Hyphal Compartment (HC)			
			N	NH_4_^+^-N	NO_3_^−^-N	^15^NH_4_^+^	^15^NO_3_^−^
			(mg L^−1^)	(mg L^−1^)		(mg HC^−1^)	
HN	AMF	NH_4_^+^	160	94	-	10	-
		NO_3_^−^	160	-	94	-	10
	NO-AMF	NH_4_^+^	160	94	-	10	-
		NO_3_^−^	160	-	94	-	10
LN	AMF	NH_4_^+^	94	94	-	10	-
		NO_3_^−^	94	-	94	-	10
	NO-AMF	NH_4_^+^	94	94	-	10	-
		NO_3_^−^	94	-	94	-	10

Note: HN means high-nitrogen treatment (160 mg L^−1^) in the root compartment, and LN means low-nitrogen treatment (94 mg L^−1^) in the root compartment.

## Data Availability

Not applicable.

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
