# Peer review of "Influence of Nitrogen Application Rate on the Importance of NO3-N and NH4+-N Transfer via Extramycelia of Arbuscular Mycorrhiza to Tomato with Expression of LeNRT2.3 and LeAMT1.1"

_plants, 2023, doi:10.3390/plants12020314_

Round 1

Reviewer 1 Report

The paper presents interesting and important topic but the experiments are not convincing for me at all. The authors used labelled compounds 15NO3 and 15NH4 but it’s not clear how 15N abundance was measured and presented. There is only mention ‘Shoot N and P concentrations were measured using a DC plasma Ecelle Spectrometer (Beckman Instruments)…’ in Material and methods. First, it should be ‘Echelle’ not ‘Ecelle’. Second, is this method able to measure nitrogen? DC plasma spectrometry is reported to measure metal elements (including P) but I am not able to find any paper/application reporting nitrogen in other sense than gas in which ionisation occurs (but argon is more typical). Even so, DC plasma spectrometry definitely will not distinguish 15N and 14N with precision necessary for this experiment. For precious isotope measurements, dedicated MS, called isotope ratio mass spectrometry (IRMS) are necessary (followed recently by various optical infrared methods).

I have also problem with your isotopic notation. What does mean 15N binding (per mills)? Is it At% of 15N? Natural abundance of 15N is about 0.37 At%, thus, your numbers are too small even for unlabelled material. Alternatively, is it delta 15N against standard (atmospheric nitrogen)? Yes, this delta is commonly reported in per mills but typical IRMS precision (StD) is about 0.2 to 0.5 per mills for nitrogen, thus, your numbers would be lower than analytical noise.

At the first look in the MS, I had have an idea ‘why the authors present only tables? Graphical presentation is commonly more readable…’ Now, I probably have a answer: ‘there are barely no significant differences in the data. In will be looking even more tristy in the plots…’

There are other formal problems to improve but in the light of my conclusion, it is certainly not necessary to mention them.

In conclusion, this MS presents not clear experiment without strong results. Thus, I cannot recommend publishing the paper in peer-review journal.

Author Response

Dear reviewer:

Thank you very much for your valuable proposal.

According to your comments, we have made changes to the following parts of the article. Please check the attached document for specific modification.

Reviewer 2 Report

The manuscript entitled “Influence of nitrogen application rate on the importance of NO3−-N and NH4+-N transfer via extramycelia of arbuscular mycorrhiza to tomato plant hosts with different expressions of LeNRT2.3 and LeAMT1.1” reports that the N transfer of extraradical mycelia is crucial for acquisition of both NO3−-N and NH4+-N by the tomato plant, especially occurred more important parts of N accumulation in plant tissue with N deficiency compared with a sufficient N. Although, the theme of the article is interesting and relevant to the journal’s scope. However, I feel numerous flaws in different sections of the ms particularly regarding methodology, data analysis and data description, as indicated below in the specific comments. A through revision/justification is required prior to publication.

-Title is confusing and too lengthy. Revise it to make it clear and catchy.. Better to report your key findings in the title.

In abstract section: Needs improvements, even treatment description is not clear. Summarize the key results. If possible, add the numerical description of most important results. Be consistent regarding units.

Introduction section contains various unnecessary statements. Remain focused on the topic. Begin with the broadest scope and get progressively narrower, leading steadily to the statement of objectives. Be clear regarding objectives.

Material and Method section: The methodology section lacks the details on experimental design.

Avoid starting a sentence with number/abbreviation. 

Discussion needs improvement. Discussion should be merely based on the observed findings. Not just a review of literature. Answer the question posed in introduction and correlate your finding with the existing knowledge.

Add a Conclusion section after discussion to report the key findings…

Check whether the format of all references is according to the journal format.

Language needs substantial improvement. There are several grammatical and typo mistakes throughout the manuscript.

All the tables and figures should be self explanatory. Define all the abbreviations in the table foot note/figure captions.

Author Response

(The authors gave the same response as above.)

Reviewer 3 Report

The manuscript “Influence of nitrogen application rate on the importance of NO3-N and NH4+-N transfer via extramycelia of arbuscular mycorrhiza to tomato plant hosts with different expressions of LeNRT2.3 and LeAMT1.1” demonstrates the importance of N transfer by extraradical mycelia of arbuscular mycorrhizal fungi for acquisition of both NO3 and NH4+ by the tomato plants. The topic of this research is actual and may be interesting for readers.

The manuscript is generally well-written. However, there are some questions that should be addressed.

Major comment

It remains unclear what the authors meant by using the term “hypersphere” (page 3, line 110).

Page 2. Using the term “Irregular Rhizobium” for arbuscular mycorrhizal fungi is not justified. Please correct this phrase.

Table 4. The capture of this table is “Effects of AMF and N form on nitrogen transporter genes expression of tomato root under different N levels”. It remains unclear from description in the text what the authors meant (the expression of genes or biomass). There is no any description about expression in the text. It should be justified in figure legend (the fold-change calculation). It should be specified in the text also.

Minor comments

Page 2, line 77. Please change the superscript for “fertilizer application”.

Page 2, line 94. Please change the superscript for “NH4+ transporter” and check it all over the manuscript.

Table 3. Specify the name “± AMF” in the table. It looks better in the Table 4. It is not clear from the table 3 which values correspond to what.

Table 5. “Effects of CAUNYN strain”. Did you mean the AMF strain CAUNYN. It should be specified.

Page 9, line 338, lime 340. Did you mean Table 4?

Author Response

(The authors gave the same response as above.)

Round 2

Reviewer 2 Report

The authors have revised the manuscript as per comments; I am happy to recommend acceptance.

Author Response

Dear reviewer:

    Thank you very much for reviewing our article.

Reviewer 3 Report

The term “hyphosphere” is more common in the literature.

Please correct the term “hypersphere” in the text.

The same names should be used in the Tables 1, 2, 3, 4 and 5 such as “High N” and “Low N” or abbreviations HN and LN.  You should present the description of HN and LN abbreviations in Figure 1 legend, if you would like to use HN and LN abbreviations only.

Please use + AMF and – AMF in the tables only, not ± AMF.  The heading ‘treatment” in the table is quite informative.

Author Response

Dear reviewer:

   Thank you very much for your valuable proposal.

   We have made changes to the following parts of the article. Please check the attached document for specific modification.
